# A Selective Histone Deacetylase Inhibitor Induces Autophagy and Cell Death via SCNN1A Downregulation in Glioblastoma Cells

**DOI:** 10.3390/cancers14184537

**Published:** 2022-09-19

**Authors:** Hui Hua Chang, Yao-Yuan Chang, Bing-Chen Tsai, Li-Jyun Chen, An-Chi Chang, Jian-Ying Chuang, Po-Wu Gean, Yuan-Shuo Hsueh

**Affiliations:** 1Institute of Clinical Pharmacy and Pharmaceutical Sciences, College of Medicine, National Cheng Kung University, Tainan 701, Taiwan; 2School of Pharmacy, College of Medicine, National Cheng Kung University, Tainan 701, Taiwan; 3Department of Pharmacy, National Cheng Kung University Hospital, College of Medicine, National Cheng Kung University, Tainan 701, Taiwan; 4Department of Pharmacy, National Cheng Kung University Hospital, Dou-Liou Branch, Yunlin 640, Taiwan; 5Department of Pharmacology, College of Medicine, National Cheng Kung University, Tainan 701, Taiwan; 6Department of Medical Science Industries, College of Health Sciences, Chang Jung Christian University, Tainan 711, Taiwan; 7International Master Program in Medical Neuroscience, College of Medical Science and Technology, Taipei Medical University, Taipei 110, Taiwan; 8Department of Biotechnology and Bioindustry Sciences, College of Bioscience and Biotechnology, National Cheng Kung University, Tainan 701, Taiwan

**Keywords:** Glioblastoma multiforme, drug resistance, selective HDAC inhibitor, LMK235, RNA-seq, SCNN1A

## Abstract

**Simple Summary:**

Glioblastoma multiforme (GBM) patients have a poor prognosis. Recent literature has shown that GBM treatment is challenging, mainly due to the heterogeneity of GBM. Epigenetic deregulation by histone modifications is found in GBM. In this study, several pan- and specific histone deacetylase inhibitors (HDACis) were evaluated for their antitumor activities in GBM cells. LMK235, an HDAC4/HDAC5 inhibitor, induced autophagy and reduced cell viability of GBM cells. Additionally, LMK235 downregulated SCNN1A expression. Downregulation of *SCNN1A* by shRNA reduced cell viability. These phenomena could be rescued by an autophagy inhibitor bafilomycin A1. Together, our findings indicated that LMK235 may be an alternative treatment for GBM cells. Furthermore, SCNN1A may be a potential therapeutic target for GBM cells.

**Abstract:**

Glioblastoma multiforme (GBM) is a grade IV, highly malignant brain tumor. Because of the heterogeneity of GBM, a multitarget drug is a rational strategy for GBM treatment. Histone deacetylase inhibitors (HDACis) regulate the expression of numerous genes involved in cell death, apoptosis, and tumorigenesis. We found that the HDAC4/HDAC5 inhibitor LMK235 at 0.5 µM significantly reduced the cell viability and colony formation of patient-derived, temozolomide-resistant GBM P#5 TMZ-R, U-87 MG, and T98G cells. Moreover, LMK235 also significantly increased TUBA acetylation, which is an indicator of HDAC inhibition. Interestingly, LMK235 induced MAP1LC3 robust readout and puncta accumulation but did not enhance PARP1 cleavage or the proportion of annexin V-positive cells, suggesting that LMK235-induced cell death occurred via autophagy activation. Further RNA-seq analysis after LMK235 treatment showed that 597 different expression genes compared to control. After bioinformatic analysis by KEGG and STRING, we focused on 34 genes and validated their mRNA expression by qPCR. Further validation showed that 2 µM LMK235 significantly reduced the mRNA and protein expression of SCNN1A. Cell viability of *SCNN1A*-silenced cells were reduced, but cells were rescued while treated with an autophagy inhibitor bafilomycin A1. Conclusively, SCNN1A plays a role in LMK235-induced autophagy and cell death in GBM cells.

## 1. Introduction

Glioblastoma multiforme (GBM) is the most common and malignant brain tumor in adults [1,2]. According to the classification of the World Health Organization (WHO), GBM is defined as grade IV astrocytoma, which has the highest malignancy and aggressiveness [3]. Although the incidence rate of GBM is relatively low, the poor overall survival and high mortality after conventional surgery, chemotherapy, and radiotherapy have attracted researchers’ attention [4,5]. Recent literature has shown that GBM treatment is challenging primarily because of its heterogeneity. A current approach from The Cancer Genome Atlas (TCGA) has revealed that GBM harbors aberrant activation of receptor tyrosine kinase/RAS/phosphoinositide 3-kinase (PI3K) signaling and downregulation of p53 signaling and Rb signaling [6,7]. Neftel et al. found that there are four states of GBM: neural progenitor-like, oligodendrocyte progenitor-like, astrocyte-like, and mesenchymal-like states [8]. The relative frequency of cells in each state varies among GBM samples and is influenced by copy number amplifications of the CDK4, EGFR, and PDGFRA loci and by mutations in the NF1 locus, which each favor a defined state.

Epigenetic deregulation by histone modifications leads to oncogene upregulation and tumor suppressor gene downregulation in numerous cancer types [9,10,11]. Aberrant epigenetic deregulation is also found in GBM, and several classes of histone deacetylases (HDACs) have been further studied [12,13]. The HDAC class I isoforms HDAC1 and HDAC2 are upregulated in GBM cell lines compared to non-neoplastic brain tissues [14,15]. Silencing HDAC1 and HDAC2 gene expression inhibits the proliferation, migration, and invasion of GBM cells. Another class I HDAC isoform, HDAC3, is overexpressed in aggressive phenotypes of glioma cell lines and is associated with a poor prognosis and overall survival of GBM patients [16]. Downregulation of the HDAC class I isoform HDAC8 leads to cell cycle arrest and O6-alkylguanine DNA alkyltransferase reduction (MGMT), which is attributed to chemoresistance in GBM cell lines [17]. Class II HDACs (HDAC4, HDAC5, HDAC6, HDAC7, HDAC9, and HDAC10) are highly overexpressed in the aggressive GBM phenotype, which has a poor prognosis [18,19,20,21]. Further study has shown that HDAC6 enhances drug resistance to temozolomide (TMZ) treatment in GBM cells [22]. Downregulation of HDAC6 expression and activity inhibits proliferation and promotes apoptosis in GBM cells. Class III HDACs include seven members of the sirtuin (SIRT) family. Some studies have shown that SIRT1 and SIRT7 are upregulated and SIRT2 and SIRT3 are downregulated in glioma cells [23,24,25,26].

Based on these findings, numerous pan-HDAC inhibitors (pan-HDACis) have been administered to treat GBM alone or in combination with chemotherapy, radiotherapy, or targeted therapy. The pan-HDACi suberoylanilide hydroxamic acid (SAHA, vorinostat) has also been combined with TMZ, radiotherapy, and other targeted therapy drugs for treatment of newly diagnosed GBM and recurrent GBM in many phase II clinical trials [27,28,29]. However, these trials did not show positive results because of ineffective outcomes and toxicity. A phase III clinical trial used the class I/II HDACi valproic acid combined with TMZ and radiotherapy to treat newly diagnosed GBM in patients without any prior treatment [30]. To minimize the toxicity of pan-HDACis and achieve better therapeutic effects, selective HDACis were identified and investigated. The class I/II HDACis valproate and sodium phenylbutyrate (PBA) were examined for their antitumor activities on commercially available GBM cell lines, such as U251-MG and GL15 [31]. Several HDAC6 inhibitors, such as tubacin, tubastatin, and J22352, were examined for their antitumor activities on U-87 MG, U251, LN299, and T98G cells [32,33,34]. However, the detailed mechanism of HDAC and related genes has not been fully clarified.

A multitarget drug against several signaling pathways is a rational and potential treatment strategy to overcome the heterogeneity of GBM. We have demonstrated that the heat shock protein 90 inhibitor NVP-AUY922 reduced oncoprotein expression and induced cell death in heterogeneous GBM cells [35]. HDACis are other multitarget drugs that regulate the expression of numerous genes involved in cell death, apoptosis, and tumorigenesis. Some HDACis have already been applied clinically, such as SAHA. However, the severe side effects caused by pan-HDACis are obstacles for their application. Therefore, the selective inhibition of HDAC is an alternative approach to minimize the side effects of HDACis and achieve antitumor effects in GBM. Further investigation of novel therapeutic targets and related drugs in GBM is crucial.

In this study, the pan-HDACi PBA and selective HDACis were used to evaluate their antitumor activities on GBM cells. LMK235, a selective HDACi, significantly reduced cell viability and colony formation of GBM cells. Cell death pathways, such as apoptosis and autophagy, were examined after LMK235 treatment. The mRNA expression alteration after LMK235 treatment was further investigated by RNA-seq and confirmed by qPCR. The data showed that SCNN1A was significantly downregulated after LMK235 treatment. Silencing *SCNN1A* reduced the viability of GBM cells. These findings indicate that HDAC4/HDAC5 inhibition is an alternative therapeutic target for GBM cells and can be achieved by LMK235 and SCNN1A.

## 2. Materials and Methods

### 2.1. Cell Culture, Chemicals, and Antibodies

GBM cells were maintained following a previous study [35]. The GBM cell lines U-87 MG and T98G were obtained from BCRC (#60360, Taiwan) and JCRB (JCRB9041, Japan), respectively. A patient-derived GBM line P#5, was obtained according to the Taipei Medical University IRB protocol (201006011). TMZ-resistant P#5 TMZ-R cells were established from P#5 cells with long-term 50 μM TMZ exposure and provided by Professor Jian Ying Chuang (Taipei Medical University, Taiwan) [36]. Cells were maintained in Dulbecco’s modified Eagle’s medium (DMEM) with 10% fetal bovine serum (FBS, HyClone, Logan, UT, USA), 100 U/mL penicillin, and 0.1 mg/mL streptomycin (GeneDirex, Taoyuan, Taiwan). All cells were maintained in a humidified incubator with 5% CO_2_ at 37 °C. LMK235 was purchased from InvivoChem. PBA, CI-994, and SW-100 were obtained from MedChemExpress. Antibodies against PARP1, MAP1LC3, and SQSTM1 were obtained from Cell Signaling Technology (Danvers, MA, USA). Antibodies against acetyl-TUBA and GAPDH were purchased from Santa Cruz Biotechnology (Dallas, TX, USA). Antibodies against SCNN1A and ACTIN were obtained from Abcam and Millipore (Burlington, MA, USA), respectively.

### 2.2. Cell Proliferation Assay

Cells were seeded in 96-well plates in at least triplicate, and after 24 h, the cells were treated with drugs as indicated dose for 7 days. At the end of the treatment, the cells were incubated with the cell proliferation agent WST-1 (Takara^®^, Shiga, Japan) for approximately 30 min. The absorbance at 440 nm was measured by a SpectraMax iD3 Microplate Reader (Molecular Devices, San Jose, CA, USA). The absorbances of all samples were normalized to the control group, in which cells were incubated without drug treatment.

### 2.3. Colony Formation Assay

Cells were seeded in 6-well plates, and after 24 h, the cells were treated with drug for 24 h. Then, the medium was replaced with growth medium, and the cells were incubated for 2 weeks. Colonies were stained with 0.5% methylene blue. The colonies were counted and normalized to the control group.

### 2.4. Immunoblotting

Cells were treated with drugs and harvested with CelLytic^TM^ M cell lysis buffer containing protease inhibitors and phosphatase inhibitors. The protein concentration was determined with the Bio-Rad^®^ protein assay dye reagent concentrate (Cat. #500-0006). Equal amounts of total protein were denatured, separated by sodium dodecyl sulfate–polyacrylamide gel electrophoresis (SDS–PAGE), and transferred to PVDF membranes. The PVDF membrane was blocked, washed, and then incubated with primary antibodies at 4 °C overnight. After incubation, the membranes were incubated with HRP-conjugated secondary antibody at room temperature. Protein bands were visualized with enhanced chemiluminescence (ECL, PerkinElmer) reagents and then captured using an iBright imaging system (Thermo Fisher Scientific^®^, Waltham, MA, USA). The density of the bands was analyzed by ImageJ software.

### 2.5. Annexin V/PI Staining

A BioLegend^®^ (San Diego, CA, USA) Annexin V-Fluorescein (FITC) Apoptosis Detection Kit with propidium iodide (PI) was used to assess cell apoptosis according to the manufacturer’s protocol. In brief, cells were seeded in 6-well plates overnight, and drug was added at the indicated dose for the indicated time. Cells were collected, washed with cold PBS, and suspended in 100 μL binding buffer containing 10 µL PI and 5 µL Annexin V-FITC. Then, the cells were incubated in the dark for 15 min and detected and analyzed by a CytoFLEX™ Flow Cytometer (Beckman Coulter Life Sciences, Lane Cove West, Australia).

### 2.6. RNA Extraction and Quantitative Analysis of mRNA

Cells were seeded in 100 mm dishes overnight and then treated with the indicated dose for the indicated time. At the end of treatment, total RNA was extracted using a Quick-RNA™ Miniprep Kit (ZYMO RESEARCH^®^, Irvine, CA, USA). RNA Lysis Buffer was added, and then the RNA was purified using Zymo-Spin™ Columns. The purified RNA was eluted by adding DEPC-treated water. The concentration of total RNA was detected using NanoDrop™ Spectrophotometers (Thermo Fisher Scientific^®^). cDNA was synthetized from 1 µg of total RNA (1 µg) by ReverTra Ace Set (PURIGO). For quantification of mRNA expression, cDNA was mixed with THUNDERBIRD^®^ SYBR^®^ qPCR Mix (TOYOBO), ROX reference dye, and primers. The qPCR assay was performed by StepOnePlus™ Real-Time PCR Systems (Thermo Fisher Scientific^®^). The primers for qPCR are listed in Appendix A. Each experiment was performed at least in triplicate, and the data are expressed as the mean ± S.E.

### 2.7. RNA Sequencing

RNA was extracted, purified, and sequenced by Genomics Inc. (Taiwan) on Illumina NovaSeq platforms. We confirmed the RNA integrity using RNA integrity number (RIN) scores (all of which exceeded 9) calculated using an Agilent 2100 Bioanalyzer. The resulting data were mapped to the human genome (hg19). Gene expression levels were quantified using RSEM software. The differentially expressed genes (DEGs) at a target FDR of 0.05 were identified by EBSeq. Gene Ontology (GO) enrichment analysis and KEGG pathway analysis were performed.

### 2.8. Lentivirus Infection

Cells at approximately 70% confluence in 6-well plates were infected with lentivirus containing pLKO.1-sh*SCNN1A* or scramble control (Sinica, Taiwan) for 24 h. Infected cells were recovered by incubation with growth media to prepare them for further assays. The effects of SCNN1A downregulation were determined by qPCR at the mRNA level and by immunoblotting at the protein level after the appropriate time.

### 2.9. Transient Transfection

Cells were transiently transfected using Lipofectamine 3000^®^ (Invitrogen) according to the manufacturer’s protocol. In brief, cells were grown to approximately 70% confluence in 6-well plates and then admixed with MAP1LC3 plasmid and Lipofectamine 3000^®^ for 6 h. Transfected cells were recovered by incubation with growth medium for 18 h and treated with LMK235 for the indicated time.

### 2.10. Statistical Analyses

We analyzed the data with the Statistical Package for the Social Sciences (Software version 16 for Windows (SPSS, Inc., Chicago, IL, USA)). All data are expressed as the mean ± SEM. (* *p* < 0.05, ** *p* < 0.01, *** *p* < 0.001). A comparison of the means among groups was performed using a one-way analysis of variance (ANOVA) followed by a Bonferroni post hoc test. The level of significance was set at *p* < 0.05.

## 3. Results

### 3.1. The HDAC4/HDAC5 Inhibitor LMK235 Reduces Viability and Colony Formation and Induces Autophagy in GBM Cells

In this study, we used GBM patient-derived, P#5 TMZ-resistant (P#5 TMZ-R) cells to evaluate the antitumor activities of selective HDACis. Two GBM cell lines, U-87 MG and T98G, were also used. PBA (Class I/II HDACi), CI-994 (class I HDACi), LMK235 (HDAC4/HDAC5 inhibitor), and SW-100 (HDAC6 inhibitor) were examined in P#5 TMZ-R, U-87 MG, and T98G cells. The dosage of each drug was based on its Cmax and the literature. PBA and LMK235 effectively inhibited the proliferation of P#5 TMZ-R cells (IC_50_ of PBA: 1449 μM; IC_50_ of LMK235: 121 nM), U-87 MG cells (IC_50_ of PBA: 2387 μM; IC_50_ of LMK235: 825 nM), and T98G cells (IC_50_ of LMK235: 443 nM) (Figure 1A–C). One micromolar CI-994 and 0.5 μM SW-100 did not inhibit the viability of P#5 TMZ-R cells. Furthermore, 0.5 μM LMK235 significantly inhibited colony formation in P#5 TMZ-R, U-87 MG, and T98G cells (Figure 1D).

Because increased acetyl-TUBA levels are an indicator of HDAC inhibition, we also checked the acetyl-TUBA level by immunoblotting to confirm HDAC inhibition by LMK235. Figure 2A and 2B shows that acetyl-TUBA was increased after LMK235 treatment in P#5 TMZ-R cells. Further data showed that LMK235 led to PARP1 reduction but no obvious PARP1 cleavage. Additionally, treatment with 0.5 µM LMK235 for 24–72 h did not increase the proportion of annexin V-positive/PI-negative cells (Figure 2C and Appendix A). Interestingly, PI-positive and annexin V-positive cells were increased, thus implying the necrosis enhancement (Appendix A). After 72 h of treatment with 0.5 µM LMK235, MAP1LC3-II conversion was a more robust readout than the control (Figure 2D,E). Furthermore, MAP1LC3B puncta were increased after 0.5 µM LMK235 treatment (Figure 2F). While cells pretreated with autophagy inhibitor bafilomycin A1, LMK235-induced cell death was significantly rescued (Figure 2G). Together, our data indicate that LMK235 reduced the viability and colony formation of GBM cells. Moreover, LMK235 treatment induced acetyl-TUBA accumulation and autophagy.

### 3.2. LMK235 Led to 597 Differentially Expressed Genes in Both 0.5 µM and 2 µM LMK235-Treated GBM Cells

Next, to verify which LMK235-mediated genes led to cell death, RNA-seq was used to measure mRNA expression after LMK235 treatment. Based on the literature and our findings, the cells were treated with 0.5 μM (low-dose) and 2 μM (high-dose) LMK235 and analyzed by RNA-seq. The transcripts were compared between the 0.5 μM LMK235-treated group and the vehicle control and between the 2 μM LMK235-treated group and the vehicle control. The volcano plot and Venn diagram of the two groups of differentially expressed genes (DEGs) (0.5 μM LMK235 vs. DMSO and 2 μM LMK235 vs. DMSO) are shown in Figure 3A,B. Compared with the vehicle control, 906 and 1736 DEGs were altered in the 0.5 μM and 2 μM LMK235-treated groups, respectively (Figure 3B). A total of 597 DEGs overlapped in the 0.5 μM and 2 μM LMK235-treated groups.

To pinpoint the important DEGs involved in LMK235-induced cell death, we focused on pathways with altered mRNA expression or genes with large fold changes. Therefore, two selection criteria for KEGG analysis and large fold changes were used. KEGG pathway analysis showed that only one pathway, the cell adhesion molecules (CAMs) pathway, was significantly altered in the 0.5 μM and 2 μM LMK235-treated groups (Figure 3C). Detailed information of KEGG analysis was listed in Appendix A. Further validation of these DEGs of cell adhesion molecules by qPCR showed that *CADM3*, *HLA-DMB*, *NFASC*, *NRXN1*, *HLA-DMA*, and *IGSF11* were significantly increased after 0.5 μM and 2 μM LMK235 treatment (Figure 3D). LMK235 also significantly reduced the mRNA expression of *LRCC4*, *CNTNAP1*, and *CLDN3*.

In addition to KEGG analysis, we also screened for genes with mRNA expression fold changes of greater than 16 or less than 1/4 and then selected the linked DEGs in the STRING database. Based on these criteria, 13 genes were downregulated and 11 genes were upregulated among 597 DEGs (Figure 4A,B). Further qPCR experiments indicated that 0.5 μM and 2 μM LMK235 treatment significantly reduced the mRNA expression of 11 of those 13 genes, *STAT5A*, *IL27RA*, *PDGFB*, *COL9A3*, *MATN1*, *SCNN1A*, *ANO1*, *RTEL1*, *GPER1*, *ADORA1*, *NOTUM*, and *FAM20C* (Figure 4C). LMK235 also increased the mRNA expression of *NTS*, *CD53*, *NCKAP1L*, *SCN1A*, *AGT*, *BEX5*, *CGA*, *NFASC*, and *CADM3* (Figure 4D).

### 3.3. LMK235 Reduces SCNN1A Expression in GBM Cells

We then tried to validate the protein expression levels of these DEGs. In this study, we aimed to find some novel DEGs by exploring the related studies in PubMed. Moreover, we searched available and reliable antibodies of these DEGs for validation by immunoblotting. Then, we picked two DEGs (LRRC4 and NRXN1) from CAM pathway, two DEGs (SCNN1A and NRXN1) from large fold changes and STRING analysis, and one DEG (CADM3) belonged to both criteria and further examined their protein expressions. Interestingly, the data showed that LMK235 only reduced SCNN1A, which was compatible with the findings from qPCR in P#5 TMZ-R cells (Figure 5A,B). Therefore, SCNN1A was chosen as a target and further validated in other GBM cells. SCNN1A is a protein unit of the epithelial sodium channel (ENaC). This channel is composed of alpha, beta, and gamma subunits and transports sodium into cells. In T98G cells, 0.5 μM and 2 μM LMK235 also upregulated the mRNA expression of *CADM3* and *NRXN1* and downregulated the mRNA expression of *LRRC4*, *NOTUM*, and *SCNN1A* (Figure 5C). The protein expression of SCNN1A was also reduced after 0.5 μM and 2 μM LMK235 treatment in T98G cells (Figure 5D,E). To investigate the role and mechanism of SCNN1A in LMK235-induced GBM cell death, shRNA against *SCNN1A* was used to downregulate the mRNA and protein expression of SCNN1A. The mRNA and protein expression levels of SCNN1A were significantly reduced in shSCNN1A-1- and shSCNN1A-2-treated cells (Figure 5F–H). Further cell function assays in SCNN1A-silenced P#5 TMZ-R and T98G cells were performed, and the cell viability of SCNN1A-silenced cells was inhibited compared with that of parental cells (Figure 5I). Furthermore, the viabilities of SCNN1A-silenced cells were significantly rescued while cells treated with autophagy inhibitor bafilomycin A1. Together, these findings indicate that SCNN1A may play a role in LMK235-inudced autophagy and cell death.

## 4. Discussion

In the present study, we found that an HDAC class II inhibitor, LMK235, significantly reduced the viability and colony formation of GBM cells. LMK235 also induced MAP1LC3 robust readout and puncta accumulation, suggesting that LMK235-induced cell death might occur via autophagy activation. Further RNA-seq analysis after LMK235 treatment showed that the levels of hundreds of mRNAs were altered. Using two selection criteria for KEGG analysis and large fold changes, we focused on 34 genes and validated their mRNA expression levels by qPCR. Further validation showed that 2 µM LMK235 significantly reduced the mRNA and protein expression of SCNN1A. Silencing *SCNN1A* expression by shRNA was confirmed by qPCR and immunoblotting against SCNN1A, and it reduced the viability of GBM cells. Conclusively, LMK235 induced autophagy and reduced cell viability by silencing *SCNN1A* expression in GBM cells. These findings on the mechanism and roles of LMK235-mediated SCNN1A expression help us better understand tumorigenesis in GBM and identify a potential therapeutic target for alternative GBM treatments.

In our study, the data showed that PBA, a pan-HDACi, and LMK235, an inhibitor of HDAC4 and HDAC5, inhibited the viability of GBM cells. CI-994, a class I HDACi, and SW-100, an HDAC6 inhibitor, did not reduce cell viability. These findings indicate that HDAC4 and HDAC5 may play a role in GBM survival and proliferation. HDAC4 has been demonstrated to be highly expressed not only in muscle and bone but also in neurons [37]. HDAC4 plays a crucial role in various brain functions, such as learning and memory [37]. In addition, a study of the tissue distribution of HDACs indicated that the expression of HDAC4 was increased in brain tumors compared to normal brain tissue [38]. Cai et al. reported that HDAC4 was upregulated in glioma tissue and glioma U251, U-87 MG, and LN-18 cells compared with glioma-adjacent normal tissues and the noncancerous glial cell line SVG p12 [18]. HDAC4 repressed the expression of CDKN1A in human cancer cells, and silencing of HDAC4 induced CDKN1A expression and inhibited cancer cell growth in vitro and in an in vivo human GBM model [39]. Research has reported that GBM patients with poor prognosis expressed higher levels of HDAC4 than patients with longer survival [40]. In addition, HDAC4 sustained the repair of DNA double-strand breaks and promoted resistance to radiotherapy in GBM cells [41]. Only a few studies have been reported on the role of HDAC5 in GBM. Gomes et al. found that HDAC5 was significantly methylated in astrocytoma compared to nonneoplastic brain samples [42]. Dali-Youcef et al. found that the expression levels of HDAC4, HDAC5, HDAC6, HDAC11, and SIRT1 were significantly and positively correlated with the survival time of patients with gliomas [43]. Therefore, we believe that HDAC4 and HDAC5 mediated gene expression are crucial for GBM tumorigenesis and proliferation.

LMK235, an inhibitor of HDAC4 and HDAC5, has been investigated for its antitumor effect on other cancers. Marek et al. previously described LMK235 as a selective HDACi against HDAC4 and HDAC5 [44]. In the human ovarian cancer cell line A2780 and its cisplatin-resistant subclone A2780 CisR, LMK235 showed the most potent cytotoxic activity, with IC_50_ values of 0.49 μM and 0.32 μM, respectively. LMK235 inhibits cell proliferation and induces apoptosis in breast cancer cell lines, diffuses large B-cell lymphoma cell lines, pancreatic neuroendocrine tumor cell lines, and multiple myeloma cell lines [45,46,47,48]. LMK235 at concentrations of 1–5 μM showed potent antimyeloma activity and downregulated heme oxygenase-1, which is an NFE2L2 transcription factor-regulated gene that may be a potential target for chemoresistant multiple myeloma treatment [49]. Additionally, LMK235 showed brain permeability, which could be important for its clinical use. After LMK235 intraperitoneal injection, histone H3 acetylation was found in the hippocampus of CDKL5 knockout mice [50]. Together, LMK235 has shown its potency for cancer treatment on GBM and other cancers. However, the detailed mechanism of LMK235-inudced cell death is not clearly investigated.

Our study attempted to explore the LMK235-mediated gene expression and revealed DEGs after LMK235 treatment on GBM cells. Further bioinformatic analysis and validation by qPCR and immunoblotting indicated that SCNN1A was reduced after LMK235 treatment. SCNN1A (sodium channel epithelial 1α subunit) is known to function in controlling electrolyte homeostasis in the body. Previous studies have revealed the correlation of SCNN1A with tumor progression. Gao et al. found that SCNN1A was overexpressed in pancreatic cancer cell lines and specimens from patients [51]. SCNN1A downregulation inhibited cell proliferation, migration and invasion, and induced cell apoptosis. Another study showed that SCNN1A upregulation by homeobox D9 (HOXD9) induced cell proliferation and migration in pancreatic cancer cells [52]. Cai et al. demonstrated that inhibition of Serine Protease 8 (PRSS8)/SCNN1A by sterol regulatory element binding protein 2 (SREBF2) reduced the cell proliferation, migration, and epithelial–mesenchymal transformation of ovarian cancer [53]. Overexpression of SCNN1A was found in ovarian cancer tissues and cell lines, and higher expression was related to poor survival [54]. Knockdown of SCNN1A by siRNA transfection reduced cell growth, invasion, and migration. SCNN1A has been reported to be the direct transcriptional target of the oncogene achaete-scute homolog 1, which can be pharmacologically targeted with antitumor effects in neuroendocrine cancer cells [55]. In breast cancer, a study has shown the potential biomarker of recurrent readthrough fusion transcripts of SCNN1A-TNFRSF1A (TNF receptor superfamily member 1A) as targets for anticancer treatment [56]. According to whole-genome expression array data for neuroblastoma (NB) cells treated with epigenetic drugs (5-Aza-dC with or without the HDACi trichostatin A) and genome-wide DNA methylation array data for NB tissue and NB cell lines, a higher methylation frequency of SCNN1A was significantly associated with poor outcome in NB [57]. Collectively, SCNN1A may play a role in proliferation and migration in cancers. Our findings also pinpoint that SCNN1A might be an alternative therapeutic target for GBM.

There were some limitations of the present study. First, we agree that there are some DEGs worth further investigation. Using RNA-seq and bioinformatic analysis as KEGG, fold changes, and STRING, we still may miss some crucial DEGs in GBM cells. Second, different doses of LMK235 treatment may lead to different patterns of DEGs. This phenomenon is also revealed in other studies and showed that LMK235 is a multitarget and multi-functional compound dependent on its dosage and cell types [58,59,60,61,62,63]. Therefore, we investigated the overlapping DEGs of 0.5 μM and 2 μM that inhibited cell viability. Third, the alternation of mRNA/protein is not always consistent. The influential factors of mRNA level include transcription, mRNA stability, RNA degradation, and miRNA. The influential factors of protein level include translation, protein stability, and degradation. Therefore, it is crucial to validate the findings from RNA-seq and qPCR by immunoblotting.

## 5. Conclusions

In conclusion, we found that the HDAC4/5 selective inhibitor LMK235 significantly reduced the viability and colony formation of P#5 TMZ-R, U-87 MG, and T98G cells. The increase in MAP1LC3 robust readout and MAP1LC3B puncta suggests that LMK235 treatment induces autophagy in P#5 TMZ-R cells. According to the RNA-seq data, we performed KEGG pathway analysis and found that the cell adhesion molecule pathway was enriched in the LMK235-treated group (both 0.5 μM and 2 μM). In total, 299 genes were upregulated and 253 genes were downregulated in the LMK235-treated group in a dose-dependent manner. Further validation shows that SCNN1A might play a role in LMK235-induced cell death of GBM cells. In conclusion, LMK235 shows potency for GBM treatment. Moreover, SCNN1A may be an alternative therapeutic target for GBM cells.

## Figures and Tables

**Figure 1 cancers-14-04537-f001:**
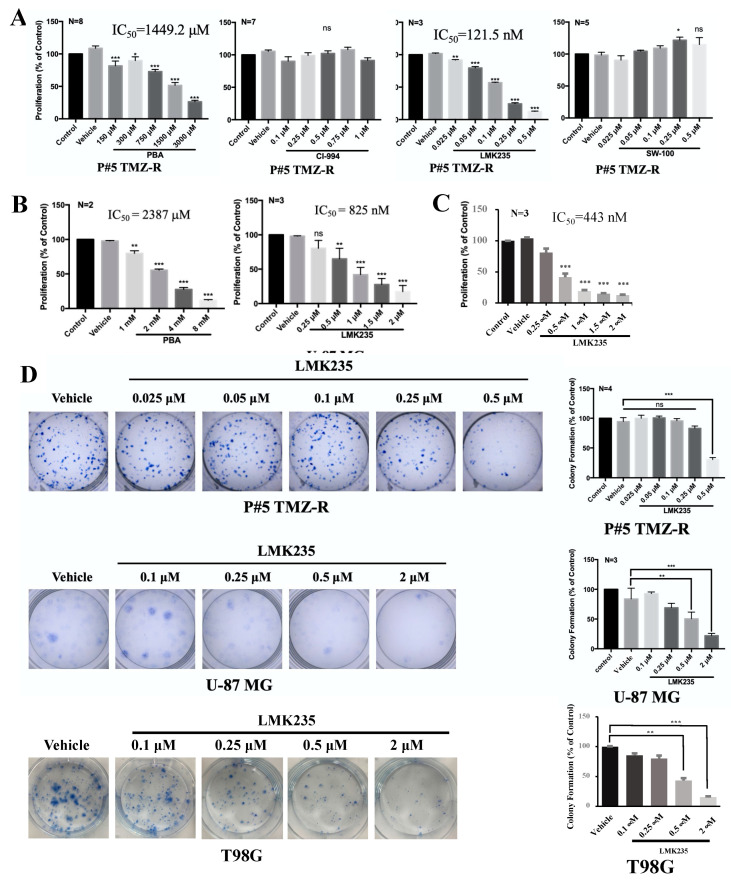
The antitumor effect of HDACi on GBM cells. Cells were incubated with PBA, CI-994, LMK235, or SW-100 at the indicated doses and then analyzed by cell viability assays (**A**–**C**) and clonogenic assays (**D**). The IC_50_ values were determined by plotting the growth relative to that of the vehicle controls. The quantification of colonies was performed manually and compared to the vehicle control. All experiments were repeated at least three times. The data are expressed as the means ± S.E. of three or more independent experiments. Scale bar = 0.5 cm in (**C**). ns: no significance. * *p* < 0.05., ** *p*< 0.01, *** *p* < 0.001.

**Figure 2 cancers-14-04537-f002:**
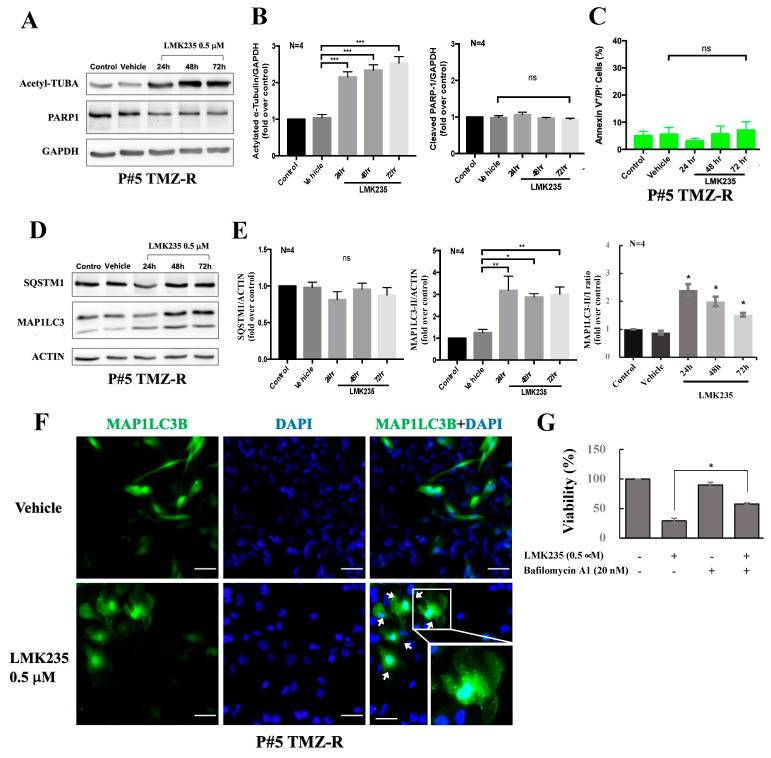
The effect of LMK235 on apoptosis and autophagy in GBM cells. Cells were incubated with LMK235 for the indicated doses and times. The cells were analyzed by immunoblotting against specific antibodies (**A**,**D**). Quantification of each band in A and D was shown in (**B**,**E**), respectively. ACTIN and GAPDH were used as internal controls. Annexin V staining was performed, and the percentage of apoptotic cells is shown in (**C**). (**F**) Cells were transiently transfected with MAP1LC3, treated with LMK235 for 72 h, fixed, and stained with DAPI. Cells were visualized by confocal microscopy, and images were acquired through the cy2 or DAPI channels (400×). The zoom-in figure was 160×. The data are representative of 5 fields/pictures of each sample. (**G**) Cells were incubated with LMK235 or bafilomycin A1 either alone or in combination at the indicated doses and then analyzed by cell viability assays. All experiments were repeated at least three times. The data are expressed as the means ± S.E. of three or more independent experiments. Scale bar = 50 μm in (**D**). ns: no significance. * *p* < 0.05., ** *p* < 0.01, *** *p* < 0.001. The uncropped blots are shown in Appendix A.

**Figure 3 cancers-14-04537-f003:**
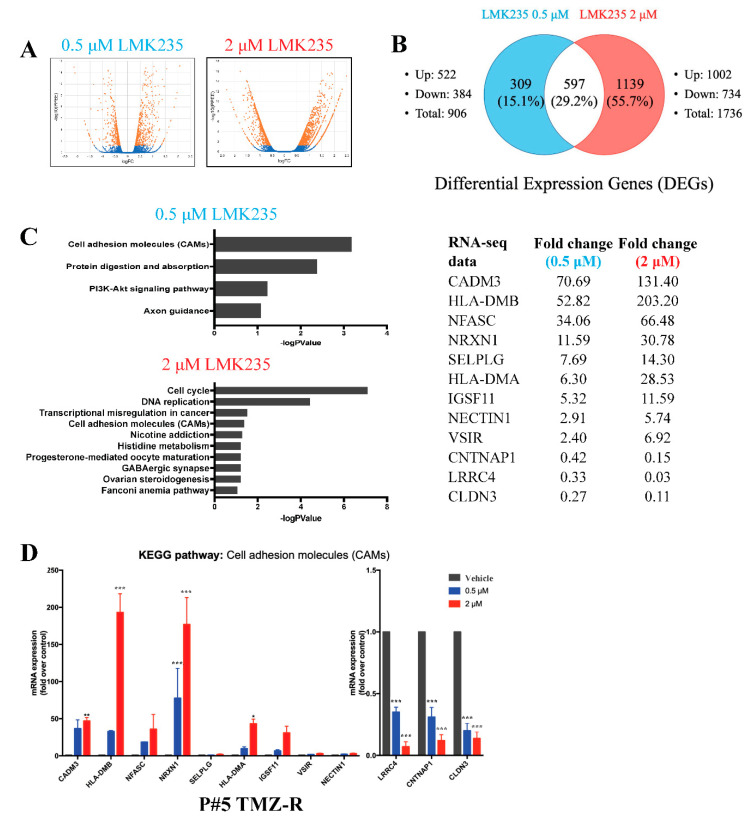
RNA-seq analysis after LMK235 treatment. Volcano plot (**A**) and Venn diagram (**B**) of DEGs in P#5 TMZ-R cells treated with 0.5 μM or 2 μM LMK235 for 72 h. (**C**) KEGG pathway analysis of the DEGs in the 0.5 μM and 2 μM LMK235-treated groups. The –logP value of the pathway in each group was ranked from highest to lowest. DEGs in the “cell adhesion molecules (CAMs)” pathway after LMK235 treatment were listed. (**D**) The mRNA expression levels of the DEGs mentioned above were verified by qPCR. * indicates comparison with vehicle; mean ± SEM. * *p* < 0.05, ** *p* < 0.01, *** *p* < 0.001.

**Figure 4 cancers-14-04537-f004:**
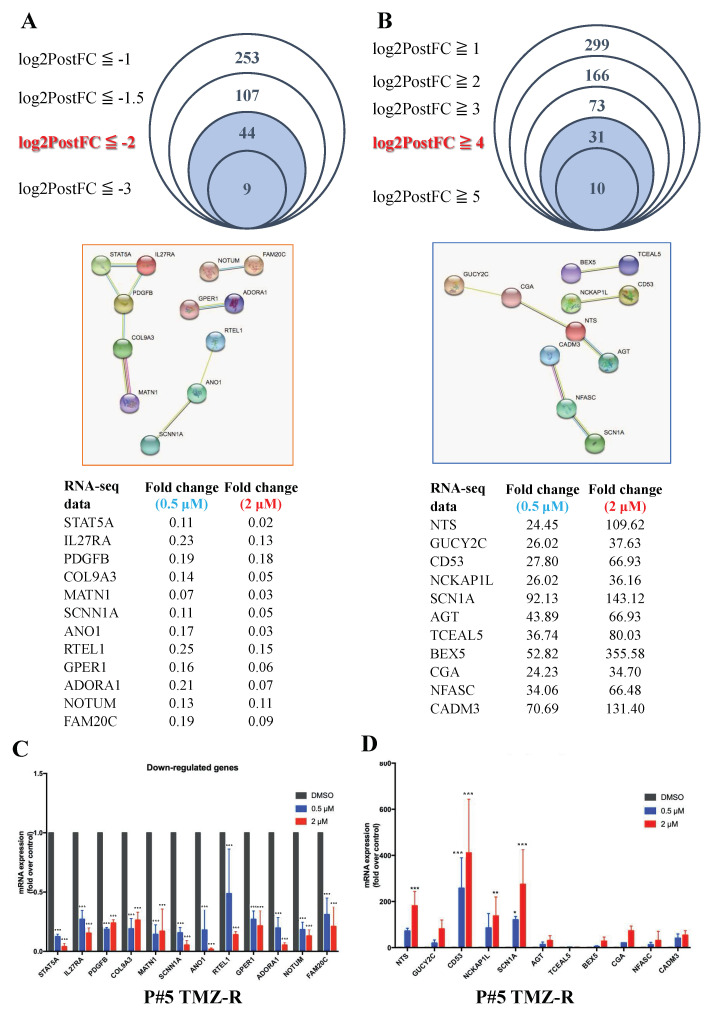
Intergroup comparison of the RNA-seq data after LMK235 treatment. (**A**) Forty-four DEGs were selected by the criterion of log2PostFC < −2, and 12 DEGs were further selected by screening their linkage to each other in the STRING database. (**B**) Thirty-one DEGs were selected by the criterion of log2PostFC > 4, and 11 DEGs were selected by screening their linkage to each other in the STRING database. The upregulated (**C**) and downregulated (**D**) mRNA expression levels of the DEGs mentioned above were verified by qPCR. * indicates comparison with vehicle; mean ± SEM. * *p* < 0.05, ** *p* < 0.01, *** *p* < 0.001.

**Figure 5 cancers-14-04537-f005:**
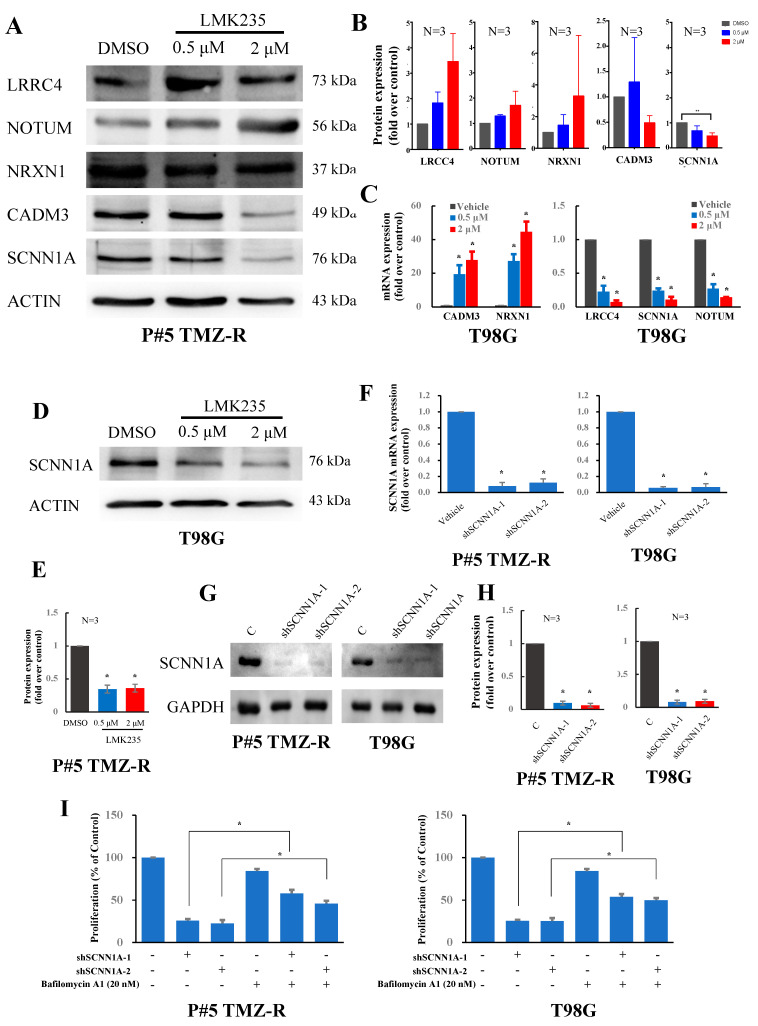
The role of SCNN1A in LMK235-treated GBM cells. (**A**) Cells were treated with LMK235 at 0.5 μM and 2 μM for 72 hr and analyzed by immunoblotting against LRRC4, NOTUM, SCNN1A, NRXN1, and CADM3. T98G cells were analyzed by qPCR (**C**) and by immunoblotting (**D**) after LMK235 treatment. The SCNN1A mRNA and protein expression levels were determined by qPCR (**F**) and immunoblotting (**G**) in *shSCNN1A*-infected cells, respectively. (**I**) *SCNN1A* silenced cells or scramble cells were incubated with bafilomycin A1 at the indicated doses and then analyzed by cell viability assays. Quantification of each band in (**A**,**D**,**G**) was shown in (**B**,**E**,**H**), respectively. ACTIN and GAPDH were used as internal controls for immunoblotting and qPCR, respectively. * indicates comparison with vehicle; mean ± SEM. * *p* < 0.05. The uncropped blots are shown in Appendix A.

## Data Availability

The Data can be shared up on request.

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
