# Peer review of "A Selective Histone Deacetylase Inhibitor Induces Autophagy and Cell Death via SCNN1A Downregulation in Glioblastoma Cells"

_cancers, 2022, doi:10.3390/cancers14184537_

Round 1

Reviewer 1 Report

This was well written paper except for minor English corrections.

Reviewer 2 Report

In the work, „A Selective Histone Deacetylase Inhibitor Induces Autophagy and Cell Death via SCNN1A Downregulation in Glioblastoma Cells“, the authors Hui Hua Chang et al. present a nice and conclusive study; it is informative, however, some aspects might be insufficient. Basically, effects of LMK 235, a HDAC inhibitor, on a glioblastoma cell line are studied.

In the work, the authors study only one single patient-derived cell line, a TMZ resistent clone, besides some commercially available long-term cultivated lines. This number is, of course, to small to draw definitive conclusions. Further, the cell line should be chaaracterized better.

As you show, low dose LMK 235 has completely different effects on gene regulation as compared with higher doses. This should be better commented on. Which concentratins might be available in vivo? Is this also seen in other HDAC inhibitors?

As far as I understand, you show that the HDAC inhibitor LMK 235 induced cell death has characteristics of autophagy and is seen in parallel with SCNN1A downregulation. But are your experiments sufficient to prove the actual sequence of events? Maybe these effects are happening just in parallel, without a causative role? Has ever been described a relationship between SCNN1A and autophagy? It would be interesting to perform an inhibtion assay with amiloride.

Unfortunately, there are several language, grammar and typing errors in the work. It should be worked over by a native speaker before publishing.

Reviewer 3 Report

In the manuscript by Chang et al., the authors analyzed several HDAC-inhibitors of which LMK235, an HDAC4/5-inhibitor, appears to be most effect against some glioblastoma cell lines for their cytotoxic effects and perform several downstream assays to further characterize the drug-induced effects.

This approach sounds reasonable and is guided by clinical trials employing HDACis as a novel treatment option for a tumor type that still represents a very dismal prognosis for the patients.

Nonetheless, this manuscript suffers from several weaknesses that should be addressed:

1. The presentation of the findings in figure 1 is very hard to crasp. Why were not all 4 drugs tested on all 3 GBM lines? This would have given a much better starting point. The data in Sub-figure A and B are very hard to crasp using the current order, labeling and presentation. This should be improved.

2. The analysis of apoptosis or autophagy are very confusing. How do the authors explain the loss of PARP1 expression after LMK235 treatment? Was a positive control employed for apoptosis determination? Regarding this assay: The presence of an Annexin V-positive can be merely considered as one piece evidence for the occurence of apoptosis. The authors should consider this. The analysis of autophagy doesn't allow for any conclusions in the present state: How do the authors explain the drop in SQSTM1 expression after 24h, whereas at later timepoints no difference can be observed. As for LC3 (Western Blot), the authors should consider calculating the ratio of the two bands. Usually an increase of the ratio is indicative of active autophagic flux. As for the IF images: These honestly don't make sense. Ongoing autophagy can be visualized by the formation of cytosolic punctae. These should also be quantified and suitable positive controls should be included as well.

3. In fact, all Western Blots should be quantified and these data should be shown next to Blots

4. The RNA-Seq appraoch is interesting, since it gives an unbiased view into the mode-of-action of the drugs. But how do the authors explain the difference (or rather the low amount of overlapping DEGs) between the two concentations. Further, the authors validated a seemingly random selection of DEGs without commenting on their biological relevance. In fact, the authors claimed to perform RNA-Seq in order to gain insight into the cell death mechanism. Were there any meaningful insights? In addition, the authors performed pathway analyses using KEGG and String using only a subset of DEGGs. Firstly, the number presented in the text are not reflected in the figures. Where are the numbers coming from? Seondly, It is much more recommended to perform pathway analyses using the entire dataset (Enrichment analyses) or using all significantly regulated DEGs. If you only feed a small dataset, you will only receive a small outcome. As for the analyzed and validated DEGs: What is their biological relevance? Why were these genes selected?

5. Lastly, the authors performed a knockdown of a seemingly randomly selected target gene. What is the rationale for choosing this gene? What is its function? Why does BafA1-treatment increase the proliferation rate of the KD-cells?

6. The discussion section also needs improvement. Currently, it is a collection of facts that are not adequately connected. Please improve.

Round 2

Reviewer 3 Report

Dear colleagues,

please find below my comments (numbering based on first review).

1. Agreed. This figure is much improved.

2. Regarding the apoptosis/autophagy discussion: To what extent is cell death induced. Based on Fig R1B is appear to be ~20% of PI-positive cells after 72h of treatment. This is a very modest increase and the question arises how biologically meaningful this even is. Additionally, considering that only a fraction of cells undergo cell death (of any kind) it is no surprise to me that no evidence for apoptosis can be found. Please comment on this.

2.1. Thank you for the clarification. Please re-consider these data in light of my comment above.

2.2/2.3 Thank you for the quantification. Generally LC-3-switch is more robust readout (not cleavage, please correct) and these data are indeed indicative of autophagy. But since cell death is likely not happening very much, the question arises about the biological function of autophagy in this setting. Is it a pro-survival mechanism as a consequence due to a toxic stimulus or an alternative pro-death mechanism?

2.4 The higher resolution picture is very helpful. I would suggest to additionally show a zoom-in to better illustrate punctae formation.

3. Agreed

4.1 I disagree. I believe it is rather that higher responses elicit the same response to a greater context and maybe as a secondary consequence activates additional processes. I would encourage the authors to thoroughly discuss this point in their discussion section. In this regard: Does it really make sense to only analyze to overlap if the authors are interesed in the cell death mechanism. Which, according to the reply, is mostly activated using the higher amount of drug?

4.2 I do not argue about the role of SCNN1A. The question is, how SCNN1A was chosen as a target. For example in Fig 3C a list of genes is presented and validated in Fig. 3D. These are profound differences and they belong to the pathway "CAM". Is this a different mechanism elicited by the drug or is this related to your protein-of-interest?

4.3/4.4 Agreed

4.5 Now I understand. This point answers my comment to 3.2. However, I cannot follow this logic in the manuscript text. I strongly advise the authors to re-phrase parts of the result section in order to make this logical connection visible to the readers more easily. Concerning the other proteins that are not regulated: Any ideas/suggestions for change in mRNA/Protein expression? Please discuss.

5. Thank you for this concise. I suggest to briefly summarize this in the result section to explain the rationale and then expand this in the discussion section. Also, I would like to point out that the data on cell death are still rather weak and the readout rather is proliferation than cell death.

6. My suggestion is to further enhance the discussion section based on the comments above.

Overall, the revised version of this manuscript is much improved compared to the previous version. Some minor adaptations still need to be made and some discrepancies need to be resolved nonetheless.
